# Multicomponent Metal Oxide- and Metal Hydroxide-Based Electrocatalysts for Alkaline Water Splitting

**DOI:** 10.3390/ma16083280

**Published:** 2023-04-21

**Authors:** Goeun Lee, Sang Eon Jun, Yujin Kim, In-Hyeok Park, Ho Won Jang, Sun Hwa Park, Ki Chang Kwon

**Affiliations:** 1Smart Device Team, Interdisciplinary Materials Measurement Institute, Korea Research Institute of Standards and Science (KRISS), Daejeon 34133, Republic of Korea; 2Graduate School of Analytical Science and Technology (GRAST), Chungnam National University, Daejeon 34134, Republic of Korea; 3Department of Materials Science and Engineering, Research Institute of Advanced Materials, Seoul National University, Seoul 08826, Republic of Korea

**Keywords:** alkaline water splitting, transition metal oxides, transition metal hydroxides, single-atom catalysts, electrocatalysts

## Abstract

Developing cost-effective, highly catalytic active, and stable electrocatalysts in alkaline electrolytes is important for the development of highly efficient anion-exchange membrane water electrolysis (AEMWE). To this end, metal oxides/hydroxides have attracted wide research interest for efficient electrocatalysts in water splitting owing to their abundance and tunable electronic properties. It is very challenging to achieve an efficient overall catalytic performance based on single metal oxide/hydroxide-based electrocatalysts due to low charge mobilities and limited stability. This review is mainly focused on the advanced strategies to synthesize the multicomponent metal oxide/hydroxide-based materials that include nanostructure engineering, heterointerface engineering, single-atom catalysts, and chemical modification. The state of the art of metal oxide/hydroxide-based heterostructures with various architectures is extensively discussed. Finally, this review provides the fundamental challenges and perspectives regarding the potential future direction of multicomponent metal oxide/hydroxide-based electrocatalysts.

## 1. Introduction

Fossil fuels have been used as an energy source by humanity since the Industrial Revolution, but the depletion of energy sources and environmental damage caused by carbon dioxide emissions have made it essential to find new, clean, and sustainable energy sources. Hydrogen, which boasts a high mass-to-ignition heat and produces no carbon dioxide emissions, is considered one of the most promising sources of clean energy [1,2,3,4,5,6]. While hydrocarbon reforming remains the primary method of hydrogen production, accounting for 95% of industrial production, it emits carbon dioxide and is therefore known as “gray hydrogen”. Electrochemical water splitting, which uses renewable energy to produce hydrogen, is viewed as a more promising technology for “green hydrogen” production due to its zero-carbon footprint, simplicity, and high energy efficiency [7,8,9,10,11,12,13].

Water electrolyzers, which split water to produce hydrogen, come in various forms including proton exchange membrane (PEM) electrolyzers and alkaline anion exchange membrane electrolyzers. PEM devices require the use of noble metal catalysts, such as Pt and Ir, and are limited by their high cost and low durability in harsh environments [14,15,16]. On the other hand, earth-abundant non-precious metal catalysts, such as Ni, Co, and Fe, have great potential for low-cost and large-scale hydrogen production when used in alkaline anion exchange membrane electrolyzers [17,18,19]. In recent years, transition metal oxides/hydroxides have emerged as efficient and stable electrocatalysts due to their high activity and stability [19,20,21]. Despite this, these catalysts face challenges such as low electrical conductivity, limited intrinsic activity, and a limited number of active sites. Multicomponent metal oxide/hydroxide electrocatalysts can overcome these limitations by combining multiple active sites and synergistic effects, resulting in improved catalytic performance. Researchers are exploring various methods to improve the catalytic properties of oxide/hydroxide-based multicomponent electrocatalysts, including interface engineering [22,23,24], alloying [25,26,27], doping [28,29,30,31], single-atom catalysts [32,33,34], and the development of phosphides [35,36], sulfides/selenides [37,38,39], and carbides/nitrides [40,41]. These methods aim to enhance the electronic structure, increase conductivity, optimize surface adsorbed species, and reduce energy barriers [42,43,44,45,46]. In particular, according to Sabatier’s principle suggesting that a suitable adsorption Gibbs free energy should be optimized for high catalytic activity, it is necessary to investigate multicomponent catalysts possessing extraordinary surface adsorption that is hard to achieve in a single metal-based catalyst. For example, single-atom catalysts enable the formation of a multicomponent catalyst to use electronic interactions with the support matrix to create new geometric structures and reduce the energy barriers of catalytic reactions [47,48,49,50]. The resulting multicomponent oxide/hydroxide-based catalysts hold great promise for future applications due to their improved reaction rates, structural/performance stability, and overall effectiveness.

Herein, we provide an overview of the surface reaction mechanism and recent progress on multicomponent transition metal-based oxide/hydroxide electrocatalysts for alkaline water splitting. First, we will introduce some basic reaction mechanisms of hydrogen evolution reaction (HER) and oxygen evolution reaction (OER) in an alkaline media. Then, the synthetic methods and characterization of the state-of-the-art electrocatalysts will be presented. Finally, the challenges and perspectives will be discussed. This review would be useful in the field of material science and chemistry for the construction and fabrication of high-performance water-splitting catalysts.

## 2. Electrochemistry of Water Splitting in an Alkaline Environment

In conventional electrolysis, the efficiency of the process depends on the concentration of charge carriers, which are typically ions in the solution. An electrolysis system and possible reaction pathways for water splitting are shown in Figure 1. Under acidic conditions, the OER at the anode corresponds to the equation 2H_2_O → O_2_ + 4H^+^ + 4e^−^. The electrons flow through the external circuit, while the protons move across a membrane to reach the cathode compartment. Then the protons combine with the electrons to form hydrogen gas (H_2_) through the HER: 4H^+^ + 4e^−^ → 2H_2_. At lower pH values, the kinetics of the water reduction is faster than in alkaline media due to the high concentration of protons (H^+^). However, at higher pH values, the concentration of hydroxide ions (OH^−^) is higher, which achieves the fast water oxidation reaction at the anode [51]. To drive the water-splitting reaction by overcoming the kinetic barriers of OER and HER, a minimum thermodynamic potential of 1.23 V is required. The role of electrocatalysts is crucial to reduce this overpotential as much as possible. Photo-electrochemical (PEC) water splitting is an attractive process as it utilizes sunlight as an energy source and emits no CO_2_ [52]. Additionally, the use of a material with a bandgap above 1.23 eV enables water splitting to be achieved without requiring an external bias. However, the property of photocatalytic water splitting is often limited by the bandgap of the photocatalytic material, which determines its absorption spectrum and energy conversion efficiency [53]. In contrast, electrocatalytic water splitting can be achieved using a variety of electrocatalysts, including metal oxides, phosphides, and sulfides. While many materials can operate as electrocatalysts, the exact mechanism by which they work is not fully understood for all materials. Thus, gaining a deep understanding of the reaction mechanism and activity relationship of HER/OER is essential in the development of efficient catalysts for electrochemical water splitting [54].

### 2.1. Mechanism of HER in Alkaline Media

In an alkaline media, the hydrogen evolution reaction (HER) is harder to achieve compared with acidic electrolytes due to the lower concentration of protons. The hydrogen intermediates (denoted as H^*^) are mainly formed by the dissociation of water molecules in alkaline media, whereas in acidic media, they are derived from hydrogen protons (H^+^) [55,56]. This means that breaking the stronger covalent H–O–H bond in alkaline media requires more energy compared with the dative covalent bond of the hydronium ion (H_3_O^+^) in acidic electrolytes. The first step of the HER process, denoted as the Volmer step, involves the adsorption of hydrogen intermediates on the surface of the electrocatalyst (Equation (1)).
M + H_2_O + e^−^ → M-H^*^ + OH^−^(1)

There are two mechanisms for forming H_2_ through either the Volmer–Heyrovsky or Volmer–Tafel step (Equations (2) and (3)).
H_2_O + M-H^*^ + e^−^ → H_2_ + OH^−^(2)
2 M-H^*^ → 2 M + H_2_(3)

A good HER catalyst must have low Gibbs free energy for H^*^ adsorption and high exchange current density with many active sites [54,57,58].

### 2.2. Mechanism of OER in Alkaline Media

There is a more complicated mechanism and slower kinetics in the OER process compared with HER, which requires a four-electron transfer process [59,60,61,62]. In alkaline media, two different pathways of O_2_ generation are possible: (i) the direct combining of adsorbed oxygen (O^*^) at two M-O^*^ species (from M-OH^*^) and (ii) the proton-coupled electron transfer (from M-OOH^*^). The hydroxide anion (OH^−^) is adsorbed on the catalyst surfaces to form M-OH^*^. M-OH^*^ is converted to M-O^*^ by coupling between H^*^ and OH^−^. Then, the combination of two M-O^*^ species directly produces an O_2_ molecule and two free M active sites. There is another pathway to generate O_2_ molecules by forming an M-OOH^*^ intermediate, which can be generated by combining M-O^*^ and OH^−^. In this pathway, an additional proton-coupled electron transfer step requires the release of an O_2_ molecule and the regeneration of the initial free active sites.
M + OH^−^ → M−OH^*^ + e^−^(4)
M−OH^*^ + OH^−^ → M−O^*^ + H_2_O + e^−^(5)
2 M−O^*^ → 2 M + O_2_(6)
M−O^*^ + OH^*^ → M−OOH^*^ + e^−^(7)
M−OOH^*^ + OH^−^ → M + O_2_ + H_2_O + e^−^(8)

The second pathway generally occurs in the OER process. Achieving the low reaction Gibbs free energy between the catalyst surface and the OER intermediates (M-O^*^ and M-OOH^*^) is important to improve the OER performance [56].

### 2.3. Strategies for Catalysts Design in Alkaline Media

Non-noble metal-based oxides/hydroxides are attractive catalysts for water electrolysis due to their abundance, low cost, adjustable structures, and stability [63,64,65,66,67,68,69,70,71,72]. However, their poor electrical conductivity and limited active sites hinder their electrocatalytic performance. To overcome these limitations, various strategies have been applied to oxide/hydroxide-based catalysts such as heterostructure engineering, doping, and anchoring single atoms. Through these approaches, the number of active sites where the reaction occurs increases, and the intrinsic conductivity of the catalysts can be improved [73,74,75,76,77,78,79,80,81,82]. The representative strategies to accomplish the advanced oxide/hydroxide-based catalysts are proposed in Figure 2.

#### 2.3.1. Heterostructures

The heterostructure, defined as a composite structure consisting of two or more different solid-state materials with interfacial interaction, has attracted attention in the field of energy-related applications due to its unique interface that provides synergistic effects such as enhanced electrochemical activity and structural stability [83,84,85,86]. As an example, Zheng’s group designed a well-defined heterointerface between Pt and LiCoO_2_ [87]. The synergistic effects at the Pt/LiCoO_2_ heterostructure interfaces enhance the Volmer step, which is a rate-determining step in the alkaline hydrogen evolution reaction (HER), by improving the capability of cleaving the HO–H bond. By optimizing the Pt–H bond affinity, fast hydrogen evolution can be achieved. In conclusion, heterostructured electrocatalysts are a promising strategy for improving intrinsic activity through the effective control of the geometric and electronic structure of the active site through interface engineering [88,89,90,91].

#### 2.3.2. Doping

Elemental doping has been considered an effective way to enhance the catalytic performance of electrocatalysts. Heteroatom doping of foreign atomic structures can alter the chemical and physical properties of electrocatalysts, resulting in improved intrinsic catalytic activity [92,93,94,95,96]. The increased surface area and additional catalytic active sites are achieved through enhanced local charge and spin density from the difference in the electronegativity and atomic radii of the anion doping elements [97,98]. There is a wealth of previous research that shows the ability of water molecule adsorption and desorption on heteroatom-doped oxides/hydroxides [99,100,101,102,103]. The intrinsic OER performance can be improved by incorporating metal cations into NiFeM (M=Co, Mn, Cr, Al, etc.) catalysts. For example, Jiang’s group found that V cations contribute to the high OER activity of NiFeOH by providing strong OOH^*^ binding energy relative to O^*^ species [98]. Van et al. successfully designed La-doped NiFe LDH using a hydrothermal method [104]. The strong electronic interactions between La and NiFe LDH elevate the Fe d-band level, increase the number of catalytic active sites and oxygen vacancies, and result in excellent OER activity. Therefore, heteroatom doping provides an ideal platform to modify various physicochemical properties such as phase transformation, vacancies, defects, and electronic band structures of oxide/hydroxide electrocatalysts [105,106,107].

#### 2.3.3. SACs

Single-atom catalysts have recently attracted enormous research interest due to their high atomic utilization efficiency, unconventional catalytic activity, and high selectivity compared with their cluster and nanoparticle counterparts [108,109,110,111,112,113,114,115,116]. Depending on the support matrix interacting with the metal single atoms, the distinct local electronic structure and coordination environment of SACs can be achieved, enabling the strong activation of the reactants. Hence, SACs are versatile for surface-active electrochemical reactions such as HER, OER, oxygen reduction reaction (ORR), CO_2_ reduction reaction (CO_2_RR), nitrogen reduction reaction (NRR), etc. [117,118,119,120,121,122,123,124,125]. In these reactions, SACs provide large amounts of active sites where the reacting elements (H^+^, OH^−^, O_2_, CO_2_, and N_2_) participate. For example, Wang et al. introduced highly dispersed Ir single atoms with a concentration of about 3.6 wt% on an Ni_2_P catalyst for OER, and they showed 28-fold higher OER efficiency compared with the most widely used IrO_2_ catalyst [126]. It was revealed that Ir single atoms preferentially occupied Ni sites of Ni_2_P, and the reorganized Ir–O–P/Ni–O–P bonding optimized the adsorption and desorption of the OER intermediate species. Consequently, the multicomponent electrocatalysts achieved by single-atomic active sites enable suppression of the intrinsic limits of metal oxide/hydroxide-derived materials by enlarging catalytically active sites with high metal utilization efficiency.

## 3. Multicomponent Oxide/Hydroxide-Based Electrocatalysts

Transition metal oxides/hydroxides, such as nickel oxide (NiO), cobalt oxide (CoO), and iron oxide (Fe_2_O_3_), have been widely studied for their potential use in the alkaline water-splitting reaction for hydrogen production [88,127,128,129]. These materials have several key properties that make them attractive for water splitting. Transition metal oxides/hydroxides showed high chemical stability in alkaline environments and excellent catalytic activity toward both HER and OER [130,131,132,133]. It is also considered a relatively low-cost catalytic material compared with novel metal ones. However, it has several limitations for following reasons: (i) the poor long-term stability due to degradation and loss of active sites in the redox process in OER, (ii) the limited stability under high overpotential conditions, and (iii) the low efficiency and high energy consumption because of the high overpotential required for the OER. Although the transition metal oxides/hydroxides show promising potential for utilization in alkaline water splitting, further research is needed to address the above limitations and improve their stability and efficiency for practical applications [134,135]. In this section, we will provide several distinct strategies to improve not only the catalytic performance in alkaline water splitting but also the long-term stability.

### 3.1. Transition Metal Oxides/Hydroxides

The advantages of heterostructure and single-atomic active sites can be simultaneously applied to the metal oxide catalysts. The heterointerface not only acts as the catalytic site where water dissociation occurs but also provides favorable anchoring sites to single-atomic metals. Zhou et al. fabricated Pt single-atom (Pt_SA_)-NiO/Ni nanosheets on Ag nanowires through a simple hydrothermal method, in which three-dimensional (3D) morphology boosted the electric conductivity and abundant active sites’ accessible channels for charge transfer [136]. The proposed HER process and water dissociation on Pt_SA_-NiO/Ni are illuminated in Figure 3a. The metallic Ni and O vacancies-modified NiO sites near Pt_SA_-NiO/Ni at the interfaces of NiO/Ni heterostructure preferred adsorption affinity toward OH^*^ and H^*^, respectively, which can be calculated by the energy barrier in the Volmer step. The high-angle annular dark-field scanning transmission electron microscopy (HAADF-STEM) image (Figure 3b) displays the immobilization of atomically dispersed Pt atoms (bright spots) along with the interfaces of the NiO/Ni heterostructure. The most of single Pt atoms are strongly anchored at the interfaces of the NiO/Ni heterostructure. Although the emission of hydrogen bubbles during the Pt_SA_ electrodeposition process can degrade the fabricated heterostructures, there are no obvious morphology changes in Pt_SA_-NiO/Ni nanosheets on Ag NWs compared with the original NiO/Ni surfaces, indicating the enhanced catalytic activity as well as high stability via the more active sites’ high structural stability. In Figure 3c, the X-ray absorption near edge structure (XANES) spectra of Pt L3-edge with each support are provided to reveal 5d occupancy of Pt since the intensity of the white-line peak indicates the transfer of the Pt 2*p*_3/2_ core electron to 5d states. From the XANES spectra, it was confirmed that the charge loss of Pt_SA_-NiO/Ni is higher than that of Pt_SA_-Ni and lower than that of Pt_SA_-NiO, as displayed in Figure 3c. From the calculated Pt oxidation states derived from the ΔXANES spectra, the average valence states of Pt were +0.29, +0.73, and +1.23 for Pt_SA_-Ni, Pt_SA_-NiO/Ni, and Pt_SA_-NiO, respectively. Charge delocalizing from Pt to the bonded O atom and charge localizing from the adjacent Ni atoms to Pt are displayed due to the different electronegativity of atoms (3.44 for the O atom, 1.91 for Ni, and 2.28 for Pt). Consequently, an enhanced electric field with a half-moon shaped area around the Pt site (Figure 3d) suggests that the NiO/Ni heterostructure coupled with a Pt single atom could possess more free electrons to promote the adsorbed H conversion for high H_2_ evolution reaction.

The Pt_SA_-NiO/Ni shows the best HER intrinsic activity compared with other reference samples (Figure 3e), a significantly lower overpotential of 26 mV at 10 mA/cm^2^, which is superior to even the commercial Pt/C catalyst (52 mV at 10 mA/cm^2^). As the advantage of single-atom catalysts, they conducted the mass activity comparison of fabricated samples. The normalized mass activity to the loaded Pt mass of the Pt_SA_-NiO/Ni sample was 20.6 A/mg at an overpotential of 100 mV, which is 2.4, 2.3, and 41.2 times greater than that of Pt_SA_-NiO (8.5 A/mg), Pt_SA_-Ni (9.0 A/mg), and the commercial Pt/C catalyst (0.5 A/mg), respectively. Furthermore, the Pt_SA_-NiO/Ni catalyst exhibited a lower Tafel slope value, 27.07 mV/dec, than the Pt_SA_-NiO (37.54 mV/dec), Pt_SA_-Ni (37.32 mV/dec), NiO/Ni (58.67 mV/dec), and Pt/C catalysts (41.69 mV/dec). In the stability test, the Pt_SA_-NiO/Ni catalysts showed high durability with negligible degradation in cyclic tests over 5000 cycles and in chronopotentiometry for 30 h. These highly durable and efficient Pt_SA_-based electrocatalysts could be achieved by anchoring on highly stable NiO/Ni heterointerfaces. From these series of theoretical and experimental results, the Pt SACs coupled with NiO/Ni heterointerfaces could boost the HER catalytic activity in an alkaline environment, leading to a significant reduction in cost. This work provides a facile way to design not only noble metal-base SACs but also the heterostructures for efficient alkaline HER.

The doping of the proper amount of noble metal can improve the catalytic activity of metal oxide catalysts. In particular, the Ru element possesses the capability to effectively dissociate H_2_O into H^+^ and OH^-^, thereby allowing for optimal hydrogen production in an alkaline condition through doping. Zhang and colleagues developed a three-dimensional (3D) needle-like array of Ru-doped Ni/Co oxides (Ru-NiO/Co_3_O_4_) based on a carbon cloth (CC) substrate via a three-step process [137]. First, Ni_0.5_Co_0.5_(OH)_2_ (NCO) nanoneedle arrays were synthesized on the surface of CC through the hydrothermal method. By controlling the Ru concentration, various Ni_0.5_Co_0.5_(OH)_2_ structures were obtained for xRu-NCO by Ru3^+^ immersion. Finally, they obtained Ru-NiO/Co_3_O_4_ nanoneedle structures after an oxidation process at 400 °C in an N_2_ environment. The morphology of the as-prepared electrocatalyst was characterized by a scanning electron microscope (SEM), as shown in Figure 4a. The lamellar morphology of Co_3_O_4_ was almost broken and cracked because the annealing treatment at high temperatures resulted in the collapse of the hydroxide structure due to the removal of water molecules, but the Ni-added NCO could maintain the needle-like array structure after oxidation. With the insertion of a high concentration of Ni content, the needle structure of NiO/CC would be denser and shorter than NCO. The staggered distribution of Co_3_O_4_ and NiO implies the creation of a large number of interfaces and high-energy regions, which can modulate the overall electronic energy state and expose a large number of active sites. The Ru 2% doped-NCO, which has the largest geometrical area with dense and uniform needle-like arrays, facilitates a sufficient contact area between the catalyst and the electrolyte, resulting in high-performance water splitting. As displayed in Figure 4a, HAADF-STEM images clearly show the presence of RuO_2_ particles with about 5 nm diameter inside needle-shaped NCOs. From the XPS O 1s core level spectra, Ru 2% doped-NCO (529.8 eV) still has the lowest anion binding energy compared with the other samples (Figure 4b) by constructing oxygen bonding between Ru atoms and oxygen vacancies. Based on these measurements, Ru doping and the NCO heterostructure system would exhibit higher catalytic activity by controlling the overall electronic energy state of the catalyst, influencing faster charge transport kinetics. From the Raman spectra, they confirmed that the heterogeneous structure of NCO and the Ru doping induces more defects. Among the various xRu-NCO samples, the 2% Ru-NCO electrocatalysts showed the best catalytic OER performance with the lowest overpotential of 233 and 269 mV at a current density of 50 and 100 mA/cm^2^ due to the densest and uniform morphology and proper distribution of RuO2 with 5 nm nanoparticles (Figure 4c). For the Tafel slope, the smallest Tafel slope value of 59 mV/dec could be obtained by fast OER kinetics originating from the fastest energy conversion and small activation energy to conduct OER. Furthermore, the 2% Ru-NCO showed excellent durability in both the cyclic test and chronopotentiometry measurement. There is a negligible potential shift in cyclic LSV tests after 1000 cycles, and only a 1.8% decayed overpotential value could be observed after 25 h of continuous operation. From the XRD and TEM analysis, the diffraction peak and crystal structure have not been changed in 2% Ru-NCO electrocatalysts after OER operation. They also investigated the HER and overall water-splitting performance of a 2% Ru-NCO electrocatalyst. It achieved a low overpotential of 51 and 138 mV at current densities of 10 and 100 mA/cm^2^, which is superior to that of Pt/C (52 and 156 mV at 10 and 100 mA/cm^2^). In the stability test, the 2% Ru-NCO showed good durability in the cyclic test over 10,000 cycles with negligible shifts and 3.4% decayed current density after 25 h of continuous operation. By utilizing 2% Ru-NCO samples as both cathode and anode in alkaline media (1 M KOH), the required potential to drive the overall water splitting up to 50 and 100 mA/cm^2^ was 1.57 V and 1.64 V with highly stable operation properties over 25 h, as shown in Figure 4d. This improved catalytic activity and stability of 2% Ru-NCO can be attributed to (i) a large number of active sites by uniformly dispersed nanostructures, (ii) low activation energy to generate NiO/Co_3_O_4_ heterointerfaces and electronic energy state modulation by Ru doping, and (iii) tight bonding between NiO/Co_3_O_4_ heterostructures and Ru dopants. This work highlights the effect of controlling the heterogeneous structures and noble Ru metal doping on water-splitting performance using metal oxide-based electrocatalysts.

Liu et al. approached two distinct strategies for synthesizing the Ce(OH)_3_-interfaced NiFe-LDH (Ce@NiFe-LDH) and the homogeneously Ce-doped NiFe-LDH (CeNiFe-LDH) catalysts on Ni foam substrate using a hydrothermal process as shown in Figure 4b and Figure 5a [138]. From the SEM measurements, the vertically and densely grown Ce@NiFe-LDH and CeNiFe-LDH could be observed with a similar nanosheet morphology of uniform thickness (~10 nm). The numerous nanoparticles could be observed in Ce(OH)_3_ decorated on the surface of NiFe-LDH, whereas a smooth surface with stacked nanosheet structures was observed in the case of CeNiFe-LDH from the HR-SEM images. The average size of the decorated nanoparticles was 7 nm without aggregation, and the (101) and (012) lattice planes could be indexed to Ce(OH)_3_ and NiFe-LDH in SAED patterns, respectively, as shown in Figure 5b. HAADF-STEM images showed the corresponding element mapping of both samples. The Ce atoms are found intensively in particle form in Ce@NiFe-LDH, but in the CeNiFe-LDH nanosheets, they are spread over entirely. Furthermore, an additional peak revealed in Ce@NiFe-LDH originated from the (101) facet of Ce(OH)_3_, indicating the coexistence of Ce(OH)_3_ and NiFe-LDH nanosheets as shown in Figure 5c from the XRD spectra. Due to the amorphous structure, both CeNiFe-LDH and NiFe-LDH exhibit no other crystalline phases, demonstrating that metal atoms are homogeneously distributed in the crystal structure of α-Ni(OH)_2_. They explained how the introduced Ce atoms could have enhanced the OER catalytic activity in terms of electronic interplay using orbital occupation between the metal and oxygen atoms, as shown in Figure 5d. The introduced Ce atoms in their structures could act as electron-accepting sites. As a result, the electron-deficient d-orbitals of Ce^3+^ strengthen the electron-accepting ability from O^2−^ to Ce^3+^, resulting in changes in the electronic structure of metal ions and influence of the OER catalytic activity in these trimetallic (oxy)hydroxides.

Ce@NiFe-LDH achieved the best OER performance with low overpotentials of 205 and 257 mV at 10 and 100 mA/cm^2^, respectively, and CeNiFe-LDH also showed good OER performances with an overpotential of 229 mV at 10 mA/cm^2^, as shown in Figure 5e. The Tafel slopes of Ce@NiFe-LDH and CeFeNi-LDH are 40.1 and 37.9 mV/dec, which can compare with that of NiFe-LDH nanosheets, originating from the increased carrier concentration and conductivity due to the presence of Fe and Ce atoms (Figure 5f). From the cyclic LSV measurement, excellent stability with negligible degradation could be observed after 48 h of operation at 1.53 V, as shown in Figure 5g. They carried out the DFT calculations to acquire the Gibbs free energies along the OER pathway at U = 1.23 V. In the atomic model, the Ni sites of NiFe-LDH, CeNiFe-LDH, and Ce@NiFe-LDH (110) surfaces were selected as the active regions on which the OH^*^, O^*^, and OOH^*^ were preferentially adsorbed. The RDS of CeNiFe-LDH was the step from OOH^*^ to O^2^ (g), whereas those of NiFe-LDH and Ce@NiFe-LDH were the step from OH^*^ to O^*^ due to the dissimilar surface charge configuration. The energy barrier of Ce@NiFe-LDH (0.56 eV) was lower than those of NiFe-LDH (0.92 eV) and CeNiFe-LDH (0.61 eV), implying that the thermodynamically favorable catalytic reaction occurs most actively in Ce@NiFe-LDH. This work demonstrated the facile route to increase the catalyst activity of NiFe LDHs by transition metal doping. By the deep understanding of OER mechanisms, the lower energy barrier in Ce@NiFe-LDH can provide not only the catalytic active sites but also stable OER performance.

Ru enables the effective dissociation of H_2_O into H^+^ and OH^−^, leading to the enhanced catalytic activity of LDHs in alkaline conditions. In recent years, single-atomic Ru has received much attention since it exhibits higher activity and atomic utilization efficiency than Ru particles due to the unique oxidation state and coordinations. In this context, incorporating single atoms of Ru into a binary CoV-layered double hydroxide (LDH) porous nanosheet array allows the design of improved electrocatalysts. Zeng et al. reported the electronic structure engineering of the binary CoV LDH porous nanosheet array with a single atom of Ru [139]. The CoV and CoVRu LDH nanosheet array was fabricated using a one-step hydrothermal method on Ni foam. A 3D interconnected hexagonal nanosheet array (about 700 nm) was synthesized uniformly on Ni foam, as seen in SEM and TEM images, as displayed in Figure 6a. The porous structure of the CoVRu LDH can be demonstrated from the evolution of NH_3_ and CO_2_ gases during the hydrothermal process, and the single atoms of Ru in their structure can be attributed to improved catalytic activity. The interplanar spacing of 0.25 nm, indexed to the (012) plane of CoV LDH, shows that the introduction of Ru does not affect the lattice spacing due to the similar ionic radius of Ru^3+^ (68 pm), V^3+^ (64 pm), and Co^2+^ (65 pm), as shown in Figure 6a. The localized electronic structures and atomic coordination of Ru atoms are revealed by XANES and X-ray absorption fine structure (EXAFS). The Ru adsorption edge (Ru K-edge) of CoVRu LDH was observed between Ru metal and RuO_2_, indicating a chemical valence state of Ru_x+_ (0 < x < 4), as shown in Figure 6b. The Fourier transform (FT) EXAFS curve of the Ru K-edge spectra of RuCoV LDH shows that the local atomic coordination of Ru is mainly from Ru–O bonds, not Ru–Ru or Ru–O–Ru bonds. These findings prove that isolated Ru single atoms form strong electronic bonds with CoV LDHs through coordination with O atoms, as illustrated in Figure 6c. The single Ru atom-anchored CoV LDH exhibited excellent HER catalytic activity and high OER catalytic performance with low overpotentials of 28 mV at 10 mA/cm^2^ for HER and 263 mV at 25 mA/cm^2^ for OER, as shown in Figure 6d. Furthermore, the CoVRu LDH had a significantly low Tafel slope of 25.4 mV/dec for HER, much lower than CoV LDH (109.3 mV/dec) and Ni foam (131.1 mV/dec), as depicted in Figure 6e. For OER, the Tafel slope was 74.5 mV/dec, indicating faster kinetics than other samples. The durability of the synthesized CoVRu LDH was evaluated with 2000 cycles for both HER and OER, showing excellent stability and maintaining the initial performance without attenuation. The overall water-splitting performance was evaluated using a CoVRu LDH || CoVRu LDH coupled electrode, requiring only 1.52 V cell voltage to achieve a current density of 10 mA/cm^2^, compared with Pt/C||RuO_2_ (1.56 V) and CoV LDH||CoV LDH (1.79 V). The favorable modulation of the electronic structure and local atomic coordination for the Ru single-atom-anchored CoV LDH electrocatalysts can be possible through DFT calculations and XPS/XAS analysis. This work suggests that improved electrocatalytic activity of Co-based LDH can be achieved by single-atom catalysts for electrochemical water splitting.

### 3.2. Transition Metal Oxide-Based Derivatives

Transition metal oxide-based derivatives, such as transition metal phosphides, sulfides/selenides, and carbides/nitrides, are widely studied as catalysts for alkaline water splitting [137,140,141,142,143]. They have advantages and disadvantages compared with traditional catalysts such as noble metals. Some of the advantages of these derivatives include lower cost, higher stability, corrosion resistance, and environmental friendliness. They are useful in a wide range of applications not only in electrocatalysis but also in energy storage and electronic devices. The durability of metal oxide/hydroxide catalysts in alkaline water splitting is an important factor in determining their potential for practical applications. When the metal oxide/hydroxide catalysts are exposed to harsh conditions, including high pH, high temperature, and corrosive electrolytes, they can be degraded over time. To enhance the durability of metal oxide/hydroxide catalysts, several strategies can be employed, such as protective coatings, the development of a more stable composition, the optimization of the reaction conditions, and chemical modifications. Furthermore, some disadvantages include lower efficiency, limited selectivity in catalytic reactions, and the need for a proper synthesis method to achieve the desired properties. Despite these disadvantages, metal oxide/hydroxide-based catalysts are still considered as one of the promising catalysts to utilized water-splitting electrocatalysts. These catalysts are still an active area of research, and further developments may lead to improved performance and greater commercial viability [144,145]. In this section, we will introduce some strategies to enhance the catalytic activity and durability of metal oxide/hydroxide-based catalysts via chemical modification.

#### 3.2.1. Transition Metal Phosphides

Transition metal phosphides (TMPs) have gained significant attention in the fields of catalysis and energy conversion due to their high catalytic activity, good thermal stability, cost-effectiveness, unique electronic properties such as high carrier mobility and high optical absorption, and high surface area [146,147]. However, several disadvantages must be addressed for the efficient use of TMPs as electrocatalysts, including poor stability in acidic conditions, limited selectivity for oxygen evolution reaction (OER), a lack of understanding of reaction mechanisms, and the cost ineffectiveness of the synthesis process [148,149,150]. In the following section, an example of designing a TMP-based electrocatalyst for overall water splitting will be provided.

Yu et al. demonstrated the efficient hybrid Fe-CoP/Ni(OH)_2_ electrocatalysts, which showed high catalytic activity in HER, OER, and overall water-splitting ability in the alkaline environment [151]. The hybrid Fe-CoP/Ni(OH)_2_ array electrode was fabricated through a coupled reaction, as shown in Figure 7a. Fe-CoP/Ni(OH)_2_ NW arrays with a few micrometer lengths have grown on the NF substrate with a corn-shaped cluster through hydrothermal synthesis. The porous Fe-CoP NW array was synthesized by the pyrolytic phosphidation process; last, the ultrathin Ni(OH)_2_ nanosheet could be electrodeposited on the outer surface of the Fe-CoP NWs. Their rational design facilitated easy access to reactants in the solution and helped the electron transfer in their heterostructures. From the TEM images in Figure 7b, the ultrathin Ni(OH)_2_ nanosheets grown on the surface and apex of Fe-CoP NWs formed a heterostructure of individual clusters. The uniform distribution of Co, Fe, and P atoms was observed in HADDF-STEM and EDS mapping images, as shown in Figure 7a. The lattice fringe with 0.29 nm corresponding to the Fe-CoP (011) facet was observed before and after Ni(OH)_2_ electrodeposition, indicating that heterostructure formation between Fe-CoP NWs and Ni(OH)_2_ was performed without structural degradation. The similar d-spacing of Fe-CoP and Ni(OH)_2_ facilitates forming their intimate contacts, which are large interfacial regions. The theoretically calculated Fe-CoP/Ni(OH)_2_ interface suggests significant charge accumulation in the interfacial region due to strong charge redistribution. The electron density increases around the interfacial Co region and depletes in the interfacial Ni region, and these electronic interactions will help to strengthen the interaction of the reactants with the catalyst surface. LDOS (black dotted line) filled with the Fe-CoP component means that it provides most of the catalytic sites. In particular, the synergistic LDOS of the Fe-CoP/Ni(OH)_2_ hybrid was significantly increased after incorporation with Ni(OH)_2_.

This efficient heterostructure clearly showed a substantially improved catalytic performance in HER and OER characterization. The overpotential of Fe-CoP/Ni(OH)_2_ was 91 mV at 10 mA/cm^2^, and the Tafel slope was 48 mV/dec in HER using 1 M KOH electrolytes, which is comparable to commercial Pt/C catalysts, as shown in Figure 6d and Figure 7c. Furthermore, the Fe-CoP/Ni(OH)_2_ hybrid electrode exhibited good durability cyclic tests over 1000 cycles and 12 h of chronoamperometry with negligible current deterioration and morphology changes. They conducted theoretical calculations to confirm the superiority of the designed hybrid catalyst in HER. From the DFT calculations, the water adsorption energy of the Fe-CoP/Ni(OH)_2_ hybrid surface (ΔG_ad_) was calculated to be −0.65 eV, which was a more negative value than that of the Fe-CoP surface (−0.5 eV), suggesting that the water molecule was more energetically favorable to adsorb on the hybrid surface, as shown in Figure 7e. Furthermore, in Figure 7f, the Fe-CoP/Ni(OH)_2_ hybrid catalyst exhibited a significantly decreased adsorption of reactant (H_2_O) on the catalyst surface (ΔG(H_2_O)) value compared with that of Fe-CoP catalyst, suggesting an effective role of Ni(OH)_2_ in promoting the dissociation of H_2_O.

For the OER, the Fe-CoP/Ni(OH)_2_ also showed excellent electrochemical properties with an overpotential of 206 mV at 10 mA/cm^2^ and a Tafel slope of 32 mV/dec, as shown in Figure 6i and Figure 7h, which is superior to the commercial OER catalyst RuO_2_. There are three aspects that are expected to improve the OER catalytic activity of the Fe-CoP/Ni(OH)_2_: (i) additional catalytic active sites from Ni(OH)_2_ nanosheets, (ii) fast charge transport due to electronic interaction between two different constituents, and (iii) lower energy barrier to adsorb the H_2_O or intermediates from interfacial edge regions. To clarify the effect of the heterostructure in OER activity, they conducted the DFT calculations with four intermediate steps. From the Gibbs free energy diagram of the Fe-CoP/Ni(OH)_2_ hybrid surface, as shown in Figure 7j, the lower ΔG(HOO^*^ = 1.77 eV) can be observed compared with that of the Fe-CoP ΔG(HOO^*^ = 2.08 eV), indicating the energetically favorable adsorption of the HOO^*^ intermediate on the hybrid surface can occur. It greatly helps to accelerate the OER kinetics, resulting in the promotion of the whole OER process. During the stability test, the hybrid electrode could maintain 92% of its initial current density at 1.5 V after 12 h. By assembling the hybrid catalysts as both cathode and anode, a two-electrode alkaline electrolyzer can be demonstrated. It exhibited the low cell voltages of 1.52 V and 1.59 V to reach the current densities of 10 and 50 mA/cm^2^, respectively, compared with the Pt/C||RuO_2_ system (1.55 and 1.66 V for 10 and 50 mA/cm^2^). These proposed hybrid catalysts shed light on the possibility to improve catalysts based on earth-abundant and non-noble transition metals for commercial water electrolysis by novel interfacial engineering.

#### 3.2.2. Transition Metal Sulfides and Selenides

Transition metal sulfides and selenides (TMSs) are a group of materials that have gained significant attention due to their unique properties and potential applications in fields such as catalysis, energy conversion and storage, and electronics [152,153]. They have high catalytic activity, good conductivity, and high stability and are low cost, making them an attractive alternative to noble metals. TMSs have good electrical conductivity and thermal and chemical stability. However, they often have poor stability in common solvents and can become less conductive at high temperatures or high humidity. The challenges in using TMSs as an efficient water-splitting electrocatalyst include the need for a facile, large-scale synthesis method with specific structures and morphologies, a better understanding of the mechanisms of TMSs in water splitting, and improved stability, especially in alkaline conditions [154,155]. The following section will introduce two examples of TMS-based electrocatalysts with improved catalytic performance and stability in alkaline water splitting.

Zhang et al. presented an interface- and defect-rich cobalt-doped Ni_3_S_2_/MoS_2_ (Co-NMS) hybridized nanosheet decorated on a hierarchical carbon framework with carbon nanowire arrays (CA) supported on conducting carbon cloth (CC). The nanosheets were prepared through a two-step process including the hydrothermal growth of the NiMoO_4_ phase and the chemical vulcanization of NiMoO_4_ at high temperatures. This two-step reaction is crucial in obtaining heterogeneous rich interfaces and small-size discrete MoS_2_ and Ni_3_S_2_ interfaces. The hydrothermally grown Co-doped NiMoO_4_ is vertically aligned on the carbon fiber with a height of 500 nm and thickness of around 50 nm. After the vulcanization at 350 °C in Ar/H_2_ (95/5 vol%), the highly segregated NiMoO_4_ nanosheets turned to Ni_3_S_2_/MoS_2_ heterointerfaces with hierarchical morphologies of nanosheets. They employed the fin-tube-like hierarchical carbon framework (CA/CC) to increase anchoring sites and to expose more surface sites, which can accelerate the mass transport (gas bubbles and electrolyte and electron transfer processes), as shown in Figure 8a [156]. From the chemical vapor vulcanization, the more reactive Ni atoms would be diffused out of NiMoO_4_ structures and react with sulfur to foam Ni_3_S_2_ particles. More particles could be precipitated out and gradually aggregated into themselves to form large particles as a function of the increase in vulcanization temperature. The aggregation of Ni_3_S_2_ particles not only destroyed the structure of nanosheets but also constructed rich and tight heterogeneous interfaces. As the vulcanization temperature increased, the diffraction peaks of MoS_2_ and Ni_3_S_2_ in NMS/CC became sharper and stronger in the XRD, indicating the improvement of the crystallinity of MoS_2_ and Ni_3_S_2_. The interlayer spacings of 0.63 nm and 0.28 nm were observed as corresponding to the (002) plane of 2H-MoS_2_ and (110) crystal plane of Ni_3_S_2_, indicating that the chemical vapor vulcanization process is suitable for generating a uniform heterogeneous interface. The tight heterointerfaces between Ni_3_S_2_ and MoS_2_, which are highlighted by the blue dotted line in Figure 8b, can provide bifunctional active sites for the cleavage of water molecules to H^+^ and OH^-^. There are many lattice defects such as dislocations, distortions, and discontinuous crystal fringes observed. These defect sites also could play the role of additional catalytic active sites for water splitting.

In the electrochemical characteristics, they conducted a variety of measurements using Ni_3_S_2_, MoS_2_, NMS, and Co-doped NMS electrocatalysts on CC and CA substrates to investigate the effect of heterogeneous interface engineering, element doping, electrode structure design, and morphology control. By the heterogeneous interface engineering, the overpotential (η10) of the Ni_3_S_2_/MoS_2_ heterostructure exhibited 146 mV at 10 mA/cm^2^, which was reduced by about 100 mV to those of one-component Ni_3_S_2_/CC (η10 = 253 mV) and MoS_2_/CC (η10 = 237 mV). The Co-NMS/CC sample showed a lower overpotential of 115 mV at 10 mA/cm^2^, and that of the Co-NMS/CA samples further decreased to 89 mV (Co-NMS/CA), which was attributed to the hierarchical carbon framework with carbon nanowire arrays, as displayed in Figure 8c. The heterointerface engineering is the most critical strategy among them, by the synergistic fast HER kinetics enabled by two components, which are Ni_3_S_2_, which acts as a water dissociation promoter, and MoS_2_, which acts as a hydrogen adsorption active site. In the Tafel slope, as shown in Figure 8d, the Ni_3_S_2_/CC sample exhibited a low Tafel slope value of 87 mV/dec, which was a remarkably reduced value compared with those of MoS_2_/CC (146 mV/dec) and Ni_3_S_2_/CC (191 mV/dec). The fastest HER kinetics in this study was observed in the Co-NMS/CC sample (62 mV/dec), which meant the interface engineering could be the most effective way to optimize the dynamic approach of HER in an alkaline solution with the largest decrease in the Tafel slope. They compared the Co-NMS/CA electrodes with the previously reported MoS_2_-based electrocatalysts in alkaline media, as shown in Figure 8e, indicating that the proposed facile strategies are effective to improve the catalytic activity of MoS_2_. Stability is another important factor to determine the efficiency of the electrocatalyst. They conducted the cyclic test and chronopotentiometry measurement at various current densities. There was a slight negative shift after 1000 CV cycles, but no obvious negative voltage shifts were found in chronopotentiometry at 10, 50, and 100 mA/cm^2^ over 50 h. This improved stability might come from the enhanced mass transport and electron transfer processes by 3D hierarchical hole structures. Furthermore, the Co-NMS/CA electrode was coupled with the NiFe LDH/CC anode to evaluate the overall water-splitting performance. The cell potential of the NiFe LDH/CC||Co-NMS/CA system is only 1.66 V at a current density of 10 mA/cm^2^, which is close to that of the NiFe LDH/CC||Pt plate system and durably operated over 10 h in alkaline media. The proposed strategy is an effective way to maximize the heterointerfaces for sulfide-based catalysts and is possibly applied to the synthesis of other heterostructures for various research fields.

Liu et al. fabricated bimetallic NiFe-based heterostructure nanosheets, consisting of both amorphous NiFe(OH)_x_ and crystalline (Ni, Fe)Se_2_ to improve the OER performance via heterointerface engineering [138]. The interface engineering modulates the electron configuration of the catalysts, resulting in enhanced electron conductivity and favorable free energies at the surface. In Figure 9a, the schematic illustration of the multistep synthesis of NiFe(OH)_x_/(Ni, Fe)Se_2_ on carbon cloth is provided. NiFe-LDH nanosheets were synthesized by the hydrothermal method followed by selenization converting them into (Ni, Fe)Se_2_/CC. Then, a core–shell structure of NiFe(OH)_x_/(Ni, Fe)Se_2_ was obtained via electrodeposition. In Figure 9b, the SEM image shows the thinly flocculent NiFe(OH)_x_ layer coated on the surface of the (Ni, Fe)Se_2_ nanosheets. Figure 9f shows the clear lattice fringe with an interplanar spacing of 0.265 corresponding to pyrite (Ni, Fe)Se_2_. In Figure 9e, the heterostructured NiFe(OH)_x_/(Ni, Fe)Se_2_ electrocatalyst exhibits excellent electrochemical OER performance with considerably low overpotentials of 180, 220, and 230 mV to achieve the current densities of 10, 100, and 300 mA cm^−2^, which are much lower than those of other electrocatalysts. Furthermore, the as-synthesized catalyst shows the Tafel slope of 42 mV dec^−1^ in Figure 9f, indicating a highly fast surface kinetics. In Figure 9g, compared with the commercial IrO_2_, RuO_2_, and recently reported OER electrocatalysts, NiFe(OH)_x_/(Ni, Fe)Se_2_ shows the lowest overpotential and Tafel slope. The authors also conducted the DFT calculation to understand the enhanced catalytic activity derived from the heterointerface between NiFeOOH and (Ni, Fe)Se_2_. In Figure 9f, the (110) facet of NiFeOOH and the (100) facet of (Ni, Fe)Se_2_ were selected considering a small interfacial strain (0.54%). The differential charge densities described in Figure 9g show that the Ni at the heterointerface has a higher chemical valence since the charge density of the Ni atom is reduced, leading to highly enhanced catalytic activity. In addition, the Gibbs energy profiles of both NiFeOOH and NiFeOOH/(Ni, Fe)Se_2_ suggest that the overpotential (0.98 V) in the rate-determining step (^*^O → ^*^OOH) of the heterostructure is much lower than that (1.1 V) in the rate-determining step (^*^OH → ^*^O) of NiFeOOH. This study not only suggested a rational design of amorphous-crystalline bimetallic heterostructures but also revealed their modified electronic coupling and their surface free energies.

#### 3.2.3. Transition Metal Carbides and Nitrides

Transition metal carbides (TMCs) and nitrides (TMNs) are interstitial compounds that embed carbon and nitrogen atoms into the interstitial sites of parent metals. These materials are unique in that they possess a combination of metallic, covalent, and ionic properties that make them ideal for various applications. They offer several beneficial properties including high electrical conductivity, hardness, corrosion resistance, thermal stability, and catalytic activity. These properties are advantageous for electrochemical water splitting. Although several TMCs and TMNs have shown promising performances, it is necessary to improve the limited surface area and difficulty in synthesis.

Yao et al. synthesized porous Cr-doped Co_4_N nanorods on carbon cloth and investigated their extraordinary electrocatalytic performance toward alkaline HER [157]. The Cr-doped Co_4_N/CC nanorod arrays were synthesized by annealing hydrothermally grown Cr-Co(OH)F with urea at 400 °C in N_2_ atmosphere. According to calculations with DFT and experimental results, it has been found that Cr atoms serve as sites for increasing water adsorption and dissociation and also modify the electronic structure of Co_4_N to enhance the hydrogen-binding capabilities of Co atoms. This leads to an acceleration in the kinetics of both the alkaline Volmer and Heyrovsky reactions. Interestingly, this approach can be applied to other metals, such as Mo, Mn, and Fe.

Diao et al. successfully synthesized heterostructured W_2_N/WC electrocatalysts to reveal the synergistic effect of the heterointerface between W_2_N and WC facilitating charge transport and separation [158]. In this research, a facile solid-state synthesis method was adapted to control the interface of the catalysts. Specifically, the porcelain boat containing blue WO_3_ powder and dicyandiamide was heated under an Ar atmosphere at 800 °C. In this condition, the W_2_N/WC heterostructure with abundant interfaces was obtained. Using DFT calculations and XAFS analysis, it was revealed that the charge density rearrangement occurred at the heterointerface, and C atoms near the interface accepted more electrons from the W atoms, resulting in electron transfer from W_2_N to WC. The resultant W_2_N/WC showed a high catalytic activity in both HER and OER with a low overpotential of 148.5 and 320 mV at 10 mA/cm^2^.

## 4. Conclusions and Outlook

Water electrolysis in alkaline conditions for AEMWE is considered the most suitable and advantageous technique for generating hydrogen energy due to the cost-effective electrocatalysts, high current density, and long-term stability. The efficacy of hydrogen generation is significantly related to the productivity of the two half-cell reactions (OER and HER) in water splitting. To minimize the overpotential required for each reaction and operate for a long time, it is critical to design highly active and robust electrocatalysts. In this respect, transition metal oxide/hydroxide-based derivatives have shown great potential since they possess a large surface area, accessible surface atoms, and tunable electronic structure. However, they have limitations in terms of low electrical conductivity and intrinsic active sites.

In this review, we summarize the superiorities of multicomponent electrocatalysts achieved by heterostructure engineering, doping, and single-atom catalysts. Heterostructure engineering induces the formation of defects and modulation of electronic configurations, leading to enhanced charge transfer and lowered surface energy. Additionally, heteroatom doping adjusts the physicochemical properties of oxide/hydroxide-based materials by modifying the morphology and acting as an electron-accepting site. Specifically, the electronic structure of metal ions in metal hydroxides is affected by the electron-deficient d-orbitals of dopants, enhancing the catalytic activity of multicomponent hydroxides. Finally, the distinct local electronic structure and coordination environment of SACs, with high atomic utilization efficiency, enables the activation of the reactants by modulating the surface energy of single-atomic sites.

Until now, a variety of oxide/hydroxide derivatives have been developed and applied to HER and OER catalysts. Among them, we have covered transition metal oxides, hydroxides, phosphides, sulfides, and selenides, to which heterostructure engineering, doping, and SACs were applied. Many researchers have strengthened the potential of oxide/hydroxide derivatives for practical utilization as electrocatalysts in the AEMWE system. Despite significant progress, there are still some issues that must be addressed for the further advance of oxide/hydroxide-based materials. We highlight three major challenges that oxides/hydroxides face.
First, oxides exhibit high durability but relatively low activity due to their highly crystalline phase and chemical stability. Hydroxides are a rising candidate with the highest activity owing to their large electrochemically active surface area, but they still suffer from low electrical conductivity. The main purposes of designing multicomponents are to increase the intrinsic activity, expose more active sites, and accelerate the electron and mass charge kinetics, improving conductivity and electrochemical performances. Thus, the innovative design and synthesis of unique nanostructures are still a great challenge in water splitting.Second, the functional roles of the active sites in oxides/hydroxides structures are not entirely clear. The nanostructured electrocatalysts undergo composition and structural transformations during the reaction under water splitting. Therefore, a deep understanding of the structural transformation is required to determine the real active phases and sites. Gaining insight into the detailed mechanism and structural transformation is critical for predicting the interaction between structure and active sites for high electrical performance in alkaline media.Finally, commercializing and simplifying the water-splitting system on a large scale needs to be further investigated to be optimized. The development of bifunctional electrocatalysis is the key factor that is active for both HER and OER reactions in the same electrolytes. Transition metal oxides/hydroxides have been reported as promising catalysts for the OER process by supporting appropriate bonding strength with adsorbed oxygen intermediates in the water-splitting process, but some catalysts are inactive in HER.

In summary, the use of oxide/hydroxide-based electrocatalysts for AEMWE has shown great potential for hydrogen production. However, there are still challenges that need to be addressed, including increasing intrinsic activity, understanding the role of active sites, and developing bifunctional electrocatalysts for both HER and OER reactions. To achieve these goals, innovative design and synthesis of unique nanostructures, a deeper understanding of the structural transformations and interactions between the structure and active sites, and the optimization of large-scale water-splitting systems are needed. Despite the challenges, the use of oxide/hydroxide-based electrocatalysts remains a promising avenue for sustainable and cost-effective hydrogen production.

## Figures and Tables

**Figure 1 materials-16-03280-f001:**
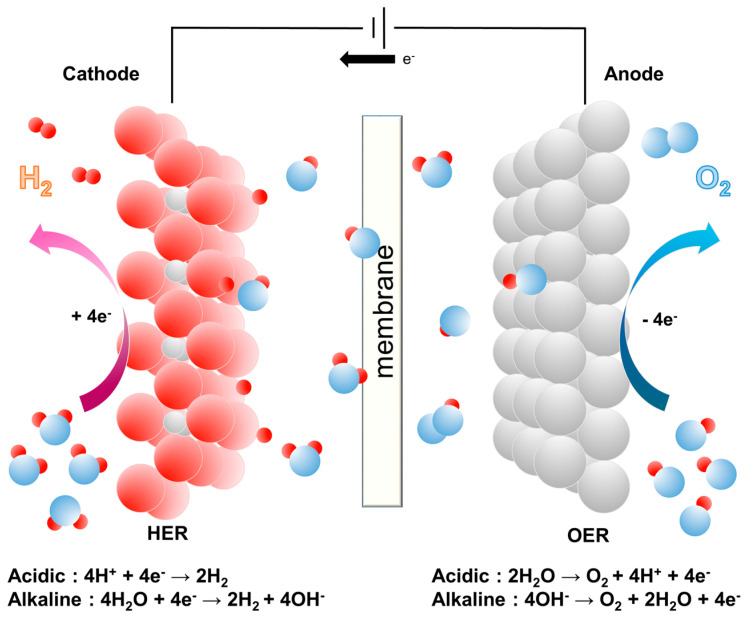
Schematic of the electrochemical water splitting and related reaction kinetics.

**Figure 2 materials-16-03280-f002:**
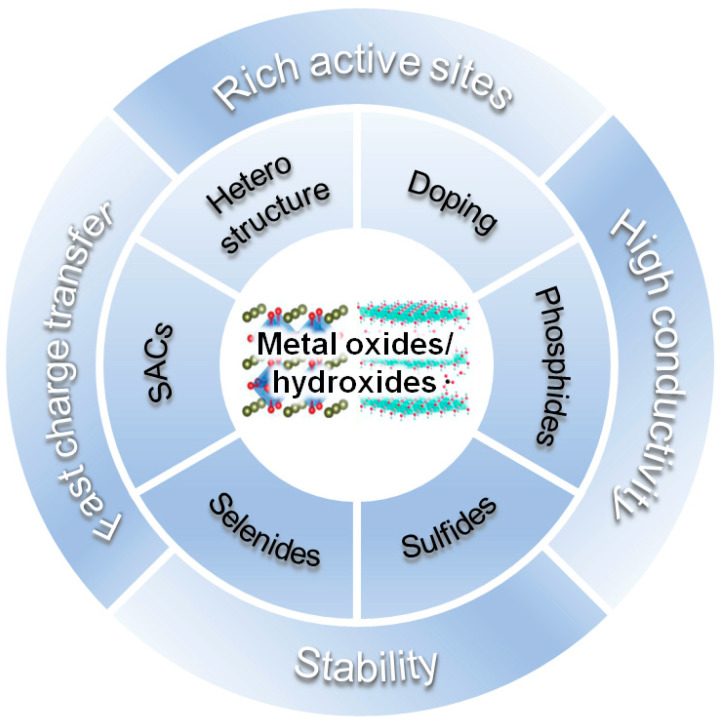
Proposed strategies to realize the improved alkaline water splitting by functionalities and advantages in multicomponent metal oxides and hydroxides.

**Figure 3 materials-16-03280-f003:**
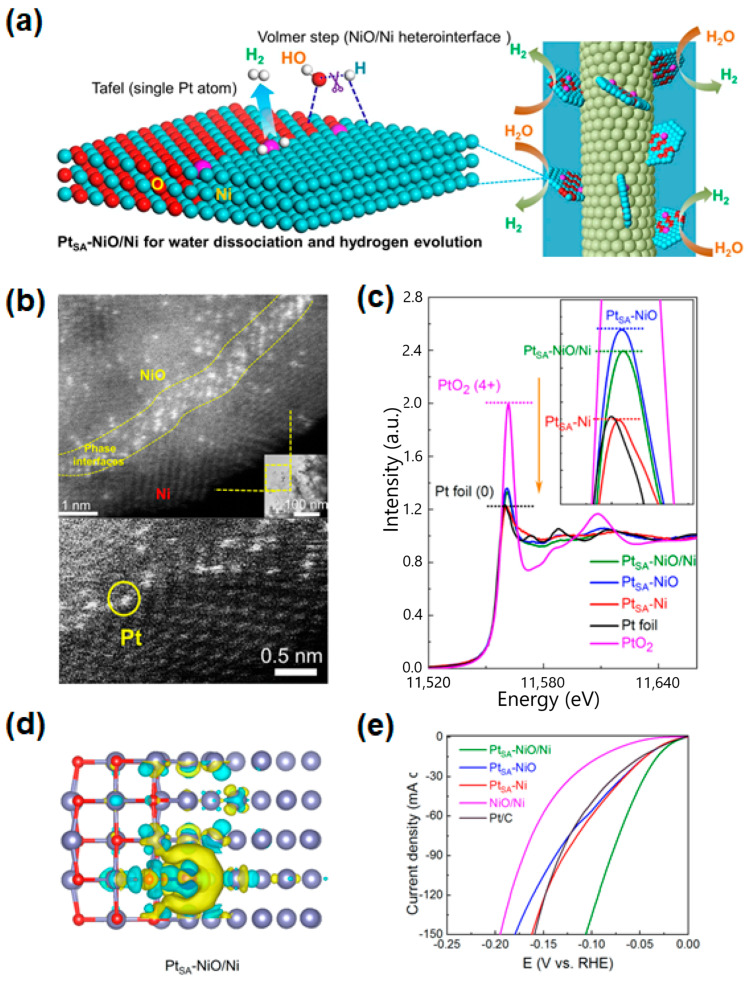
(**a**) The mechanism of the Pt_SA_-NiO/Ni network as an efficient catalyst toward large-scale water electrolysis in alkaline media. (**b**) HAADF-STEM image of Pt_SA_-NiO/Ni. (**c**) XANES spectra and calculated Pt oxidation states derived from ΔXANES spectra of Pt_SA_-NiO/Ni, Pt_SA_-NiO, and Pt_SA_-Ni, with Pt foil given as a reference. (**d**) Computational models and localized electric field distribution of a Pt_SA_-NiO/Ni. (**e**) LSV curves of Pt_SA_-NiO/Ni, Pt_SA_-NiO, Pt_SA_-Ni, NiO/Ni, and Pt/C for HER. Reprinted (adapted) from reference [136], copyright (2021) Springer Nature.

**Figure 4 materials-16-03280-f004:**
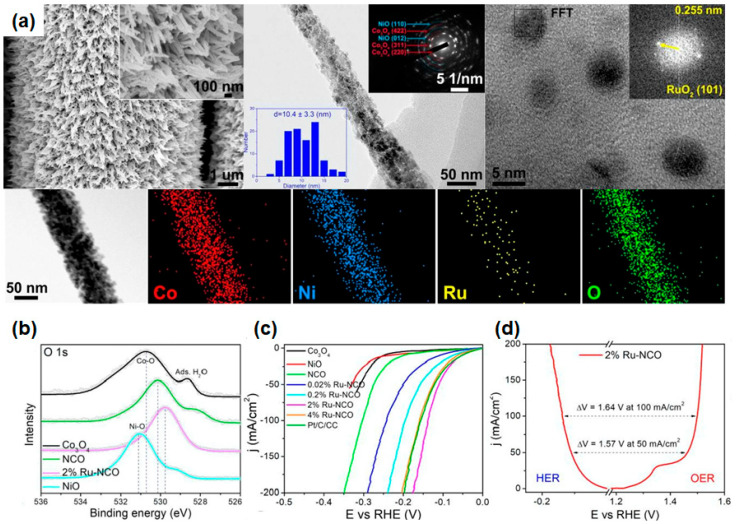
(**a**) (left) SEM and (middle) TEM images of 2% Ru-NCO with high magnification images. The corresponding SAED pattern and histogram of particle size distribution are also presented. (right) HR-TEM image of RuO_2_ particles with the corresponding SAED pattern of the selected area. Elemental mapping images for Co, Ni, Ru, and O (below). (**b**) XPS patterns for Co_3_O_4_, NCO, 2% Ru-NCO, and NiO in O 1 s core level spectra. (**c**) HER LSV curves of Co_3_O_4_, NiO, NCO, 0.02% Ru-NCO, 0.2% Ru-NCO, 2% Ru-NCO, 4% Ru-NCO, and RuO_2_/CC in 1.0 M KOH. (**d**) LSV curves of 2% Ru-NCO for HER and OER in a three-electrode configuration. Reprinted (adapted) from reference [137], copyright (2022) Elsevier B.V.

**Figure 5 materials-16-03280-f005:**
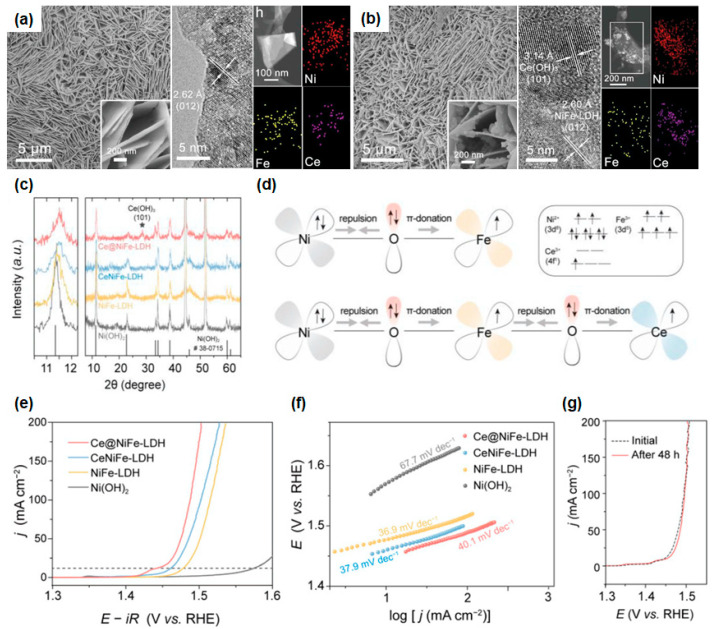
SEM, high-resolution TEM, HAADF-STEM images, and the corresponding EDX elemental mappings of (**a**) CeNiFe-LDH and (**b**) Ce@NiFe-LDH nanosheets. (**c**) XRD patterns of as-prepared Ce@NiFe-LDH and CeNiFe-LDH. The data for Ni(OH)_2_, NiFe-LDH, and Ce(OH)_3_ are shown for comparison. (**d**) Schematics of the electronic interplay among Ni, Fe, Ce, and O in NiFe-LDH, CeNiFe-LDH, and Ce@NiFe-LDH. (**e**) LSV curves. (**f**) The corresponding Tafel plots of Ce@NiFe-LDH, CeNiFe-LDH, NiFe-LDH, and Ni(OH)_2_ for OER. Measured O_2_ yields at 0.3 V. (**g**) LSV curves of Ce@NiFe-LDH before and after 48 h electrolysis. Reprinted (adapted) from reference [138], copyright (2021) Wiley-VCH GmbH.

**Figure 6 materials-16-03280-f006:**
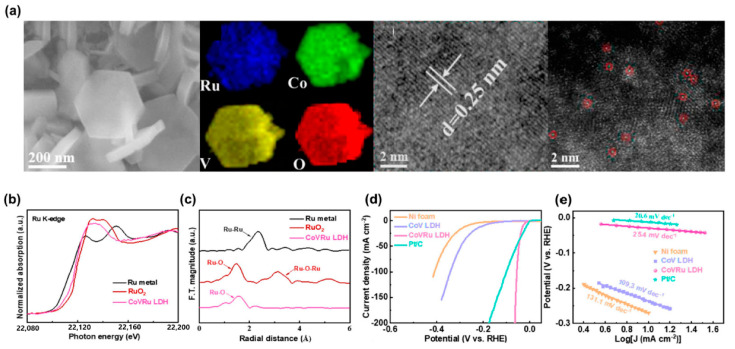
(**a**) SEM image (left) of hexagonal nanoplatelets of synthesized CoVRu LDH. STEM elemental mapping images with Ru, Co, V, and O. HR-TEM and HAADF-STEM images with d-spacing of 0.25 nm and atomically dispersed Ru sites, which are highlighted by red circles in CoVRu LDH. (**b**) Normalized XANES spectra and (**c**) Fourier-transformed EXAFS spectra of Ru K-edge for Ru metal, RuO_2_, and CoVRu LDH. (**d**) HER polarization curves and (**e**) calculated Tafel slope graph. Reprinted (adapted) from reference [139], copyright (2023) Elsevier B.V.

**Figure 7 materials-16-03280-f007:**
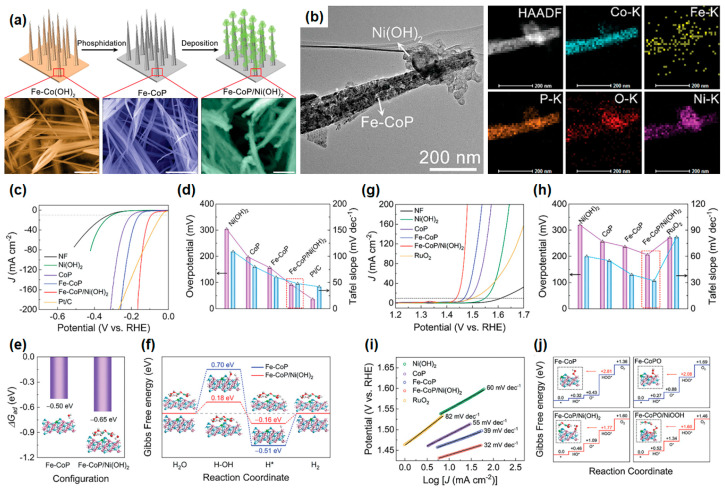
(**a**) Schematic illustration of the synthesis steps of the Fe-CoP/Ni(OH)_2_ hybrid and SEM images of Fe-Co(OH)_2_, Fe-CoP, and Fe-CoP/Ni(OH)_2_ nanowires. (**b**) TEM image, HAADF-STEM image, and EDS mapping images of the Fe-CoP/Ni(OH)_2_ hybrid nanowire. (**c**) The iR-corrected LSV curves and (**d**) comparison of overpotential and Tafel slope at different electrodes for HER in 1 M KOH. (**e**) Comparison of calculated water adsorption energy (ΔGad) and (**f**) calculated Gibbs free energy diagrams for alkaline HER at the bare Fe-CoP surface and the Fe-CoP/Ni(OH)_2_ hybrid surface. (**g**) iR-corrected LSV curves, (**h**) Tafel plots, and (**i**) comparison of overpotential and Tafel slope at different electrodes for OER in 1 M KOH. (**j**) Calculated Gibbs free energy diagrams of the OER pathway at the Fe-CoP, Fe-CoP/Ni(OH)_2_, Fe-CoPO, and Fe-CoPO/NiOOH surfaces. Reprinted (adapted) from reference [151], copyright (2021) Wiley-VCH GmbH.

**Figure 8 materials-16-03280-f008:**
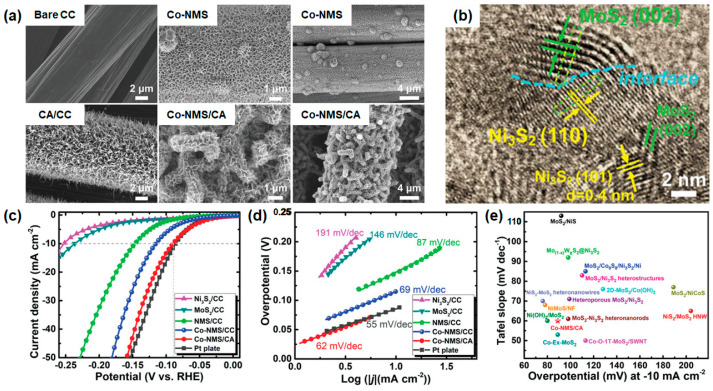
(**a**) SEM images of bare CC, Co-NMS/CC, CA/CC, and Co-NMS/CA. (**b**) HRTEM image of Co-NMS. (**c**) LSV curves of HER for various electrocatalyst samples and (**d**) corresponding Tafel slopes. (**e**) Comparison of the overpotential at −10 mA/cm^2^ and Tafel slope on various representative MoS_2_-based HER electrocatalysts in alkaline electrolyte. Reprinted (adapted) from reference [156], copyright (2021) Wiley-VCH GmbH.

**Figure 9 materials-16-03280-f009:**
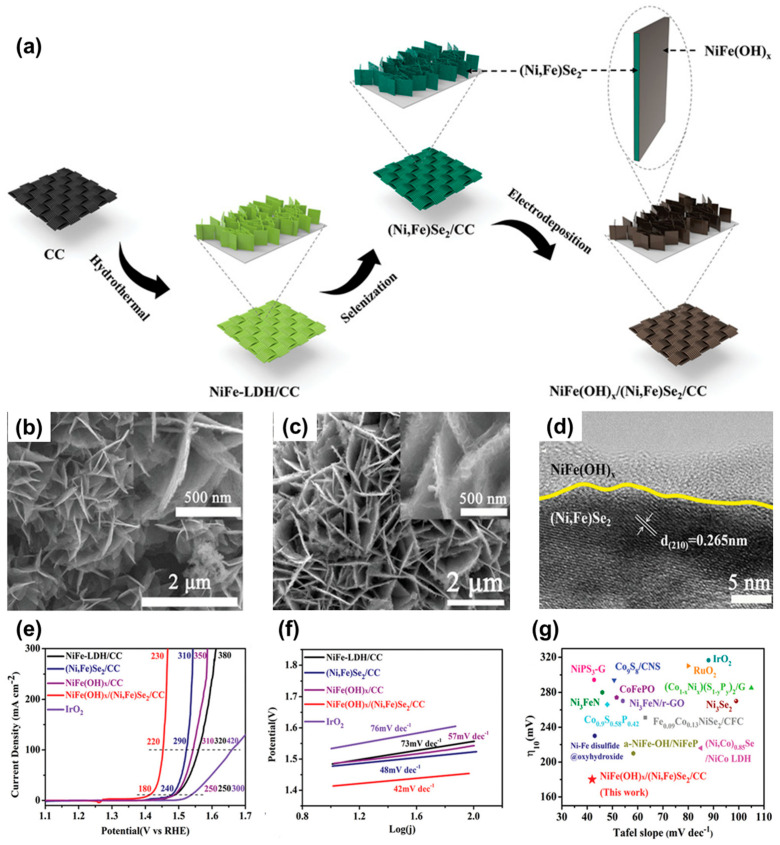
(**a**) Schematic illustration of the synthesis of NiFe(OH)_x_/(Ni, Fe)Se_2_ on carbon cloth. SEM images of (**b**) (Ni, Fe)Se_2_/CC, (**c**) NiFe(OH)_x_/(Ni, Fe)Se_2_/CC, and (**d**) HRTEM images of NiFe(OH)x/(Ni, Fe)Se_2_/CC. (**e**) LSV curves and (**f**) Tafel plots of IrO_2_/CC, NiFe-LDH/CC, (Ni, Fe)Se_2_/CC, NiFe(OH)_x_/CC, and NiFe(OH)_x_/(Ni, Fe)Se_2_/CC. (**g**) Overpotential required at 10 mA/cm^2^ (η_10_) and Tafel slope comparison of the catalysts in this work with other reported high-performance OER electrocatalysts. Reprinted (adapted) from reference [138], copyright (2021) Wiley-VCH GmbH.

## Data Availability

Not applicable.

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
