# Peer review of "Multicomponent Metal Oxide- and Metal Hydroxide-Based Electrocatalysts for Alkaline Water Splitting"

_materials, 2023, doi:10.3390/ma16083280_

Round 1

Reviewer 1 Report

This manuscript summarizes the recent advances in developing advanced strategies to synthesize multicomponent metal oxides/hydroxide-based materials, such as nanostructuring, heterointerface engineering, single-atom catalysts, and chemical modification. This manuscript is well organized. I think it can be considered for publication after minor revisions as listed below.

1.      The authors should describe the motivation for designing multicomponent metal oxides/hydroxide materials in the introduction.

2.      In “3.2 Transition metal oxide-based derivatives”, the authors introduce transition metal phosphides, sulfides, and selenides. What about other derivatives including carbides and nitrides? Can the authors briefly discuss the progress of these materials?

3.      On page 4 line 111, two many references #71-97 are inserted in the same place. I suggest the authors carefully remove and separate some of these references.

4.      On page 11 line 394, the authors claimed higher stability of the derivatives than metal oxides/hydroxides. I think the authors should refer to “durability”. Details should be inferred.

5.      The resolution of Figure 4 and 5 should be improved. 

Author Response

April 10, 2023

Dear Editor,

We have revised our manuscript entitled Multicomponent metal oxides/hydroxides-based electrocatalysts for alkaline water splitting (Manuscript ID: materials-2269561) which had been submitted for publication in Materials.

We have elaborately revised the manuscript according to the reviewers’ comments. We attach the point-by-point response to the reviewers’ comments, and the revised manuscript with the changes highlighted in blue.

Thank you for spending your time and I am looking forward to hearing from you.

Sincerely yours,

Ki Chang Kwon, Ph.D.

Reviewer 2 Report

In this review, Lee et al. described “Multicomponent metal oxides/hydroxides-based electrocatalysts for alkaline water splitting” in detail. All the sections are explained very well. However, at this stage there are still many problems and I therefore suggest a major review for this review article keeping in mind the following questions.

1) The English language is very poor and the article needs proper attention to attract more reviewers.

2) No correlation exists between different parts of the review. For example, in section 2.3. “Strategies for catalysts design” and then no strategy has been described, similarly, “Unique advantages of oxides/hydroxides for alkaline HER” is just followed by “Heterostructures” and “doping”. My question is that first described some properties, some advantages and disadvantages and then add suggestion for example doping of the pure materials, heterostructure formation with other materials etc.

3) The respected authors just copied the work of other researchers and removed plagiarism. No attention has been given how to start the paragraph. For review writing, generally a paragraph is started with some basic ideas and then work of other authors is incorporated and highlighted that these particular authors have work under these ideas and improved the efficiency of the catalysts.

4) The figures are very blur, and the respected authors are requested to add high resolution figures.

5) Some very important citations are missing.

i) T. Ilyas, F. Raziq, N. Ilyas, L. Yang, S. Ali, A. Zada, S.H. Bakhtiar, Y. Wang, H. Shen, L. Qiao, FeNi@CNS nanocomposite as an efficient electrochemical catalyst for N2-to-NH3 conversion under ambient conditions, J. Mater. Sci. Technol. 103 (2022) 59-66.

ii) T. Ilyas, F. Raziq, S. Ali, A. Zada, N. Ilyas, R. Shah, Y. Wang, L. Qiao, Facile synthesis of MoS2/Cu as trifunctional catalyst for electrochemical overall water splitting and photocatalytic CO2 conversion, Mater. Des. 204 (2021) 109674.

iii) A. Zada, M. Khan, M. N. Qureshi, S. Liu, R. Wang, Accelerating photocatalytic hydrogen production and pollutant degradation by functionalizing g-C3N4 with SnO2, Front. Chem. 7 (2020) 941.

iv) A. Zada, M. Khan, Z. Hussain, M. I. A. Shah, M. Ateeq, M. Ullah, N. Ali, S. Shaheen, H. Yasmeen, S. N. A. Shah, A. Dang, Extended visible light driven photocatalytic hydrogen generation by electron induction from g-C3N4 nanosheets to ZnO through the proper heterojunction, Z. Phys. Chem. 236 (2022) 53-66.

6) No schematic diagrams are seen in the review. Can the respected authors provide figures with schematic diagram showing charge separation and HER in meaningful ways?

7 Can the author provide a paragraph showing comparison of the acidic and alkaline water splitting, photocatalytic and electrocatalytic water splitting? This will significantly improve the quality of the review article.

Author Response

(The authors gave the same response as above.)

Round 2

Reviewer 2 Report

Since the authors made significant changes and improved the quality of the paper by responding well to all my questions and carrying out necessary changes, I therefore accept the publication of this paper in your reputed journal.